# Syngas as Electron Donor for Sulfate and Thiosulfate Reducing Haloalkaliphilic Microorganisms in a Gas-Lift Bioreactor

**DOI:** 10.3390/microorganisms8091451

**Published:** 2020-09-22

**Authors:** Caroline M. Plugge, João A. B. Sousa, Stephan Christel, Mark Dopson, Martijn F. M. Bijmans, Alfons J. M. Stams, Martijn Diender

**Affiliations:** 1Laboratory of Microbiology, Wageningen University & Research, Stippeneng 4, 6708 WE Wageningen, The Netherlands; joaoasousa@gmail.com (J.A.B.S.); fons.stams@wur.nl (A.J.M.S.); martijn.diender@wur.nl (M.D.); 2Wetsus, European Centre of Excellence for Sustainable Water Technology, Oostergoweg 9, 8911 MA Leeuwarden, The Netherlands; martijn.bijmans@wetsus.nl; 3Centre for Ecology and Evolution in Microbial Model Systems (EEMiS), Linnaeus University, SE-391 82 Kalmar, Sweden; mail@stephanchristel.de (S.C.); mark.dopson@lnu.se (M.D.)

**Keywords:** acetogenesis, biomass retention, carbon monoxide, carrier material, formate, haloalkaline biodesulfurization, hydrogen, sulfate reduction, thiosulfate reduction

## Abstract

Biodesulfurization processes remove toxic and corrosive hydrogen sulfide from gas streams (e.g., natural gas, biogas, or syngas). To improve the efficiency of these processes under haloalkaline conditions, a sulfate and thiosulfate reduction step can be included. The use of H_2_/CO mixtures (as in syngas) instead of pure H_2_ was tested to investigate the potential cost reduction of the electron donor required. Syngas is produced in the gas-reforming process and consists mainly of H_2_, carbon monoxide (CO), and carbon dioxide (CO_2_). Purification of syngas to obtain pure H_2_ implies higher costs because of additional post-treatment. Therefore, the use of syngas has merit in the biodesulfurization process. Initially, CO inhibited hydrogen-dependent sulfate reduction. However, after 30 days the biomass was adapted and both H_2_ and CO were used as electron donors. First, formate was produced, followed by sulfate and thiosulfate reduction, and later in the reactor run acetate and methane were detected. Sulfide production rates with sulfate and thiosulfate after adaptation were comparable with previously described rates with only hydrogen. The addition of CO marginally affected the microbial community in which *Tindallia* sp. was dominant. Over time, acetate production increased and acetogenesis became the dominant process in the bioreactor. Around 50% of H_2_/CO was converted to acetate. Acetate supported biomass growth and higher biomass concentrations were reached compared to bioreactors without CO feed. Finally, CO addition resulted in the formation of small, compact microbial aggregates. This suggests that CO or syngas can be used to stimulate aggregation in haloalkaline biodesulfurization systems.

## 1. Introduction

Hydrogen sulfide is a toxic and corrosive compound present in gas streams such as natural gas or biogas. In biodesulfurization systems, sulfide is removed from the gas and elemental sulfur is formed. Biodesulfurization systems that produce elemental sulfur operated under haloalkaline conditions have been studied and applied for more than 25 years [1,2]. The efficiency of converting sulfide to elemental sulfur in laboratory experiments is close to 97% [2] and models suggest that 98% efficiency is possible [3,4]. The remaining part of the sulfide is biologically oxidized to sulfate or chemically to thiosulfate. These compounds are soluble and accumulate in the system causing a pH decrease that has a negative impact because high pH (>8.5) is required for efficient hydrogen sulfide gas absorption. To solve this problem, part of the liquid content of the system is removed, generating a bleed stream with high pH and high salinity and containing sulfate and thiosulfate [1].

This bleed stream can be treated by biological reduction of sulfate and thiosulfate to sulfide in a separate bioreactor. This allows the recirculation of the bleed stream into the sulfide-oxidizing bioreactor of the biodesulfurization process [5,6]. Previous studies showed that this is possible under haloalkaline conditions in different bioreactor types using a variety of electron donors [7,8,9]. Hydrogen gas (H_2_) is considered a suitable electron donor for biodesulfurization systems applied for natural gas as it can be produced on-site by gas reforming of methane. However, syngas is produced in the gas reforming process that consists mainly of H_2_, carbon monoxide (CO), and carbon dioxide (CO_2_). Purification of syngas to obtain pure H_2_ implies higher costs because of additional post-treatment. The application of syngas as an electron donor without removal of CO is thus preferred and was previously applied successfully at neutral pH and low salinity conditions [10]. On the other hand, CO is known to be toxic to microorganisms [11,12]. Because of its strong reductive properties (E^0^ = −520 mV), CO reacts with metals in the active centres of enzymes [13]. Via this mechanism CO is known to inhibit both [Fe-Fe] and [Ni-Fe] hydrogenases [14] of which the former appears more sensitive to inhibition by CO [15,16]. Presence of CO can therefor inhibit H_2_-driven sulfate reduction, and other potential hydrogen-dependent metabolisms such as formate, acetate, and methane production. 

Despite its toxicity, several types of microbial metabolism are able to use CO as substrate. Acetogens are relatively well-known CO-utilizing microorganisms, and their use for production of fuels and chemicals from syngas is currently considered at an industrial scale [17]. CO-driven metabolism is also found in sulfate reducers and methanogens [11,12,18]. Sulfate reducers are mainly found to convert CO via the water-gas shift reaction to produce H_2_ (Table 1, Equation (1)), that is subsequently used as electron donor for sulfate reduction [19,20,21]. Previously, it was shown that sulfate-reducing biomass from bioreactors at neutral and high temperature conditions could be adapted to CO as substrate [18,22]. Additionally, it was observed that CO had an effect on the formation of granules, making them smoother [22]. This was probably caused by the development of different layers on the granule, with a potential external layer containing acetogenic bacteria that convert CO while the inner layer contained sulfate-reducing bacteria. These sulfate-reducing bacteria likely used the produced acetate as an additional carbon source [22]. Granules with a smoother shape settle better than irregular shaped aggregates, and in this way CO could induce the formation of aggregates with enhanced settling properties.

To date, the effect of CO on haloalkaphilic microorganisms has not been studied in pure or mixed cultures. In this study, we investigated syngas as an electron donor for a sulfate and thiosulfate-reducing bioreactor operated under haloalkaline conditions. The impact of CO on sulfate and thiosulfate reduction as well as on formate, acetate, and methane production was investigated. In addition, biomass attachment to sand as well as microbial aggregation was studied in order to understand if CO can be used to influence the adhesion of microorganisms similar to what was observed at neutral pH and low salinity conditions [22]. The feasibility of industrial application of syngas as electron donor for sulfate and thiosulfate reducing bioreactors was evaluated and the main advantages and disadvantages of the use of syngas versus pure hydrogen gas are discussed.
CO + 2H_2_O → HCO_3_**^−^** + H_2_ + H^+^(1)
4CO + 4H_2_O → C_2_H_3_O_2_**^−^** + 2HCO_3_**^−^** + 3H^+^(2)
4H_2_ + 2HCO_3_**^−^** + H^+^ → C_2_H_3_O_2_**^−^** + 4H_2_O(3)
4CO + SO_4_^2**−**^ + 4H_2_O → 4HCO_3_**^−^** + HS**^−^** + 3H^+^(4)
4H_2_ + SO_4_^2**−**^ + H^+^ → HS**^−^** + 4H_2_O(5)
CO + H_2_O → HCO_2_**^−^** + H^+^(6)
H_2_ + HCO_3_**^−^** → HCO_2_**^−^** + H_2_O(7)
4CO + 5H_2_O → CH_4_ + 3HCO_3_**^−^** + 3H^+^(8)
4H_2_ + HCO_3_**^−^** + H^+^ → CH_4_ + 3H_2_O(9)

## 2. Materials and Methods

### 2.1. Bioreactor Set-Up

A 4.4-L glass gas lift reactor with an internal three phase separator was used [7] (Appendix A). The temperature was maintained at 35 °C using a water jacket connected to a thermostat bath (DC10-P5/U, Haake, Dreieich, Germany). The influent feeding was performed by a membrane pump (Stepdos 08 RC, KNF-Verder, Utrecht, The Netherlands). The H_2_, CO, and CO_2_ gas supply was controlled using digital mass flow controllers (F-201CV-020-AGD-22-V and F-201CV-020-AGD-22-Z, Bronkhorst, Ruurlo, the Netherlands). The gas was recycled using a vacuum pump (Laboport^®^, KNF, Trenton, NJ, USA) and the gas flow was measured using a calibrated flow meter (URM, Kobold, Arnhem, the Netherlands). A pH and a redox potential sensor (CPS11D and CPS12D, Endress + Hauser, Naarden, the Netherlands) connected to a controller (Liquiline CM44x, Endress + Hauser, Naarden, the Netherlands) were used to monitor the conditions inside the reactor. The pH was controlled at pH 9 by supplying CO_2_ via the mass flow controller. As biomass carrier material, 0.5 L of acid washed sea sand with a particle size of 0.1–0.3 mm (VWR, Amsterdam, The Netherlands) was added as described in [23]. 

### 2.2. Inoculum

The inoculum was a 1/1 ratio mixture from a sulfate and thiosulfate reducing gas-lift bioreactors fed with H_2_ and CO_2_ [7,8,23]. For the inoculation, 100 mL of concentrated biomass from each bioreactor was used, obtained by centrifuging the content of the previously operated bioreactors at the end of the experiments. 

### 2.3. Medium

A mineral medium was used that was buffered with sodium carbonate and sodium bicarbonate at pH 9 ± 0.05, that contained a total of 1.5 M Na^+^. The medium composition was as follows: Na_2_CO_3_ (33.6 g L^−1^), NaHCO_3_ (69.3 g L^−1^), KHCO_3_ (1 g L^−1^), K_2_HPO_4_ (1 g L^−1^), NH_4_Cl (0.27 g L^−1^), MgCl_2_.6H_2_O (0.1 g L^−1^), CaCl_2_.2H_2_O (0.01 g L^−1^), and 10 mL L^−1^ of vitamin solution [24]. Two trace element solutions were added [7]. As electron acceptors, 7.1 g (25 mM) of sodium sulfate and 3.95 g (12.5 mM) of sodium thiosulfate were added.

### 2.4. Experimental Design

After filling it with medium, the bioreactor was flushed with H_2_ gas overnight to lower the redox potential. The gas recirculation was set at 5 L min^−1^ (±0.5). Then the H_2_ gas supply was set at 20 mL.min^−1^ and the pH control (set at 9) was turned on. The inoculum was added (time 0) that initiated the start-up phase. Hereafter, the only parameter changed was the gas composition by adding CO (Table 2). 

On day 217 a CO spike experiment was performed, where the CO supply was increased to 60% of the gas phase and maintained for 48 h before being switched to 15% on day 219 until the end of the experiment.

### 2.5. Batch Experiments

Two batch experiments were performed to assess the metabolic capacity of the biomass at different stages of the bioreactor operation. All experiments were performed with the same medium composition used for the bioreactor. 

To study the activity of the biomass after CO addition, 125 mL of biomass containing bioreactor liquid from day 54 of operation was transferred to 250 mL serum bottles. The serum bottles headspace was replaced with N_2_ gas. CO was added to make three different gas compositions: 30, 55, and 80% CO (at 1 bar of total pressure). The bottles were incubated at 37 °C and shaking at 150 rpm for 14 days. Each CO condition was tested in triplicate. For each sampling point, 5 mL of liquid content and 1 mL of gas content were collected from the bottles. 

To study the effect of formate and acetate on sulfate/thiosulfate reduction in adapted biomass in the presence of CO, 120 mL serum bottles were filled with 70 mL fresh medium and N_2_ headspace. Three different conditions were tested: no electron donor, with 25 mM sodium formate, and with 12.5 mM sodium acetate. Biomass containing bioreactor liquid (50 mL) was collected at day 90, centrifuged for 10 min at 10,000× *g*, re-suspended with 10 mL buffer (pH 9, 1.5 M Na^+^), and added as inoculum. CO was added at approximately 25% CO in the gas phase (1.4 bar total pressure). Each set of batch cultures was tested in triplicate. For each sampling point, 2 mL of liquid content and 1 mL of gas content were collected from the bottles.

### 2.6. Analytical Procedures

Liquid samples for volatile fatty acids, sulfate, thiosulfate, and sulfide analysis were prepared and analyzed as described previously [7]. The H_2_, CO, CO_2_, N_2_, and CH_4_ in the gas phase were quantified by gas chromatography using a CP-4900 microGC (Varian, Palo Alto, CA) as previously described [7]. 

To measure the biomass concentration, the sand-attached biomass was separated from the suspended biomass by settling for 30 s and both fractions were transferred to a new separate tube. One milliliter of sand was collected and the samples were washed three times with a carbonate/bicarbonate buffer with lower salinity (LS buffer; pH 9, 0.5 M Na^+^ instead of 1.5 M Na^+^). In subsequent washing steps, sand was separated from the buffer by 30 s settling. Two milliliter of the suspended biomass was centrifuged (10 min, 10,000× *g*) and washed three times with LS buffer. Finally, the total nitrogen content was determined using a cuvette test (LCK238, Hach Lange, Düsseldorf, Germany).

The particle size of the bioreactor content (including sand) was measured using laser measurement in a particle size and shape analyzer (Eyetech, Doner technologies, Or Akiva, Israel) with the Dipa 2000software (Doner technologies, Or Akiva, Israel). Each sample was analyzed in triplicate and each measurement was performed continuously for 120 s with stirring. Microscopy pictures were taken using a light microscope (DMI6000B, Leica, Biberach, Germany). The sand particles were not measured because of settling in the mixing chamber, being only small aggregates analyzed for particle size.

### 2.7. Scanning Electron Microscopy

Biomass samples were fixed in 2.5% (*w*/*v*) glutaraldehyde overnight at 4 °C. The fixed samples were separated and washed following the same procedure for attached fraction and suspended fraction described above for biomass measurement. Then the samples were dehydrated in a series of ethanol solutions, (10%, 25%, 50%, 75%, 90% and twice with 100%) with 20 min in each step and then dried in a desiccator. The samples were coated with gold and analyzed in a JEOL JSM-6480LV scanning electron microscope (JEOL Benelux, Nieuw-Vennep, The Netherlands). 

### 2.8. DNA Isolation and Bacteria Community Profiling

Samples (10 mL) for DNA analysis were collected on days 0, 46, 96, 123, 216, and 218 of bioreactor operation and stored at −80 °C. The samples were separated into the attached and suspended fractions and washed following the same protocol as for biomass measurements. Total genomic DNA from the suspended fraction of all samples and attached fraction of samples from day 123 was extracted using the PowerBiofilm™ DNA Isolation Kit (MoBio, Carlsbad, CA, USA) following the manufacturer’s instructions. DNA was stored at −20 °C.

A fragment of the 16S rRNA gene of bacteria, including the V3-V5 regions, was amplified with primers 341F and 805R [25]. Sequences were submitted to the ENA database (http://www.ebi.ac.uk/ena) under the accession number PRJEB11708. The PCR protocol and sequencing using the Illumina Miseq platform were performed as previously described at the Science for Life Laboratory, Sweden (www.scilifelab.se) [26,27]. The sequencing data were processed with the UPARSE pipeline and annotated against the SINA/SILVA database (SILVA 119) [28,29]. Finally, the data were analyzed using Explicet 2.10.5 [30]. 

### 2.9. Calculations

Calculations and assumptions:

Thermodynamic calculations under actual bioreactor conditions were performed using the online tool eQuilibrator 2.0 (equilibrator.weizmann.ac.il) [31]. The relative electron donor use calculations were based on the molar quantities indicated by the reactions described in Table 1. The assumptions used in the calculations were as follows: (i) Evaporation does not cause a major loss of liquid from the bioreactor because it is a closed system; (ii) the bioreactor liquid volume was assumed to be constant during the operation meaning that the liquid flow going into the bioreactor was equal to the liquid flowing out of the bioreactor (Q_in_ = Q_out_); (iii) accumulation of sulfur compounds by incorporation in biomass and formation of sulfide precipitates was assumed to play a negligible role. This was due to the high sulfate and thiosulfate concentration in the influent compared to the low concentration of metals added to the medium; (iv) hydrogen used for biomass synthesis was assumed to play a negligible role in the bioreactor; (v) the N fraction value of 0.2 was used to calculate biomass concentration based on total N, following the biomass molecular formula: C_1_H_1.8_O_0.5_N_0.2_; and (vi) the attached biomass fraction was composed of the sand particles with attached microorganisms and microbial aggregates without support material.
$$C_{x}=\frac{C_{TotalN}}{0,2}.M_{x}$$
$$C_{tx}=C_{x,sp}+C_{x,a}$$
$$r_{vs}=\frac{Q.C_{s,in}-Q.C_{s,out}}{V_{r}}$$
$$r_{vp}=\frac{Q.C_{p,out}-Q.C_{p,in}}{V_{r}}$$

## 3. Results and Discussion

### 3.1. Bioreactor Performance

The use of syngas as an electron donor for haloalkaliphilic sulfate and thiosulfate reducing microorganisms was possible up to 15% CO. However, a period of adaptation of the biomass to CO was required. All H_2_-driven microbial processes, including sulfate/thiosulfate reduction and formate production, were inhibited by the presence of 5% CO (Figure 1A,B). This is in agreement with similar studies performed at neutral conditions where the sulfate reduction activity decreased from 140 mmol L^−1^ d^−1^ to 98 mmol L^−1^ d^−1^ with 5% CO [22]. During this inhibition period (day 47 to 75), thiosulfate disproportionation to sulfate and sulfide was not inhibited. 

This can be observed by the increase of sulfate and sulfide concentrations in an approximately 1:1 ratio (Figure 1). During this period, in which no acetate was formed, the pH decreased and the CO_2_ fraction in the gas phase increased that was indicative of the water-gas shift reaction (Equation (1), Table 1, and Figure 1C). After 38 days of operation with 5% CO (day 84), the sulfate/thiosulfate reduction activity recovered. This activity was not further affected by an increase to 15% CO on day 124 (Figure 1A). Thiosulfate was completely reduced while 87 ± 3% of the sulfate was reduced during the stable phase with 15% CO (day 200 to 216). The sulfidogenic rate was similar to that achieved in a study with only H_2_ (42.3 ± 2.2 mmol_S_ L^−1^ d^−1^), when applying the same sulfate/thiosulfate loading rate [23]. The sulfidogenic rates (r_vs_) achieved almost matched the loading rates of sulfate and thiosulfate (Figure 2A). This indicated that sulfidogenic rates could be higher if loading of sulfate and thiosulfate was increased as described for other sulfate and thiosulfate reducing bioreactors (Table 3). Moreover, increasing the length of the different stages may contribute to a further increase in the sulfidogenic rates.

Besides sulfate and thiosulfate reduction, acetate, formate, and methane were produced (Figure 1a,b and Figure 2b). Acetate production started after the inhibition period on day 76, probably using hydrogen and/or CO as electron donors. It was not possible to distinguish if either H_2_ or CO were specifically used as H_2_ and CO were simultaneously consumed (Figure 2c). However, looking at the overall electron donor consumption (H_2_ + CO), acetate production was the main electron sink in the system. Approximately 49% (±5%) of the supplied electron donors in the stable phase with 15% CO was converted to acetate (Figure 3). This was considerably higher than reported in bioreactors fed with H_2_ or formate as electron donors [9,23]. Furthermore, high acetate concentrations might have contributed to increased biomass growth compared to previous results (Table 3, Figure 4). This effect of acetate was observed when no other carbon sources were added to the media [23]. 

Even though high and stable sulfate and thiosulfate reduction rates were achieved with 15% CO in the gas supply (Figure 1 and Figure 2), temporary CO increases could still have a negative and disturbing effect on the bioreactor operation. To study this, 60% CO was fed to the bioreactor from day 217 until day 219 before returning the CO to 15%. The addition of 60% CO led to a steep decrease in sulfate reduction, formate and acetate production, and biomass concentration (Figure 1 and Figure 4). However, thiosulfate disproportionation was not affected and methane production even increased after the 60% CO spike (Figure 1 and Figure 2). This indicated that either methanogenesis could compete for H_2_ after inhibition of the other hydrogenotrophic microorganisms or that the methanogens present were using CO as the electron donor. Generally, methanogens are rapidly inhibited upon CO exposure, and are considered more sensitive to CO compared to acetogens [32,33]. However, methanogenesis with CO has never been reported under haloalkaline conditions and it is thermodynamically favorable at the conditions present in the bioreactor (Table 1). 

### 3.2. Inhibition by CO 

As observed in previous studies, hydrogen-driven metabolism can be rapidly inactivated upon CO exposure, and is usually thought to be related to hydrogenase inhibition [34,35]. The results here show that upon exposure of the biomass to CO, H_2_-driven sulfate reduction activity dropped rapidly, as well as formate and acetate production (Figure 1). This is potentially related to the inhibition of hydrogenases. The observation that thiosulfate disproportionation remained active supports this hypothesis, as thiosulfate disproportionation does not require the action of hydrogenases. However, some hydrogenases have been reported to be highly resistant to CO like some [Ni-Fe]-hydrogenases and a single [Fe-Fe] hydrogenase [14,36,37,38]. Microorganisms capable of producing such CO-resistant hydrogenases would be able to metabolize H_2_ in CO-rich environments. Some hydrogenogenic organisms operating the water gas-shift reaction employ such hydrogenases [37], explaining potential hydrogenogenic activity during the CO-inhibited phase.

Despite the seeming inhibition of many hydrogenase-dependent conversions, biomass levels remained relatively constant during this phase, indicating microbial growth (Figure 4). Besides thiosulfate disproportionation and the water-gas shift reaction, acetate production from CO might also have occurred during this phase. Even though acetate was under the detection limit, its production in lower amounts might have supported the biomass growth. Slow oxidation of CO was observed in batch bottles incubated with biomass from the period just after CO was added (Figure 5). This suggests that the microbial community present during this operation period had the ability to convert CO. 

### 3.3. Adaptation to CO 

After 30 days without activity of H_2_-driven microbial processes, at day 76 acetogenic activity was observed that correlated with the removal of CO from the headspace (Figure 1 and Figure 2). Several acetogens are able to utilize CO as substrate via the Wood-Ljungdahl pathway [34]. In volcanic environments that contain high levels of CO, carboxydotrophic organisms play a role in the removal of CO, thereby creating a viable environment for non-CO-tolerant microbes [39]. The CO-driven acetogenic activity observed in the bioreactor could have created favorable conditions for hydrogenotrophic sulfate reducers and formate producers. Additionally, adaptation to CO by microorganisms already present in the biomass, such as via production of CO-resistant enzymes, also might have contributed to the restoration of activity. Some hydrogenases have been reported to be highly resistant to CO like some [Ni-Fe]-hydrogenases and O_2_ tolerant hydrogenases [14,36,37]. Microorganisms capable of producing such CO-resistant hydrogenases would be able to metabolize H_2_ in CO-rich environments. 

The effect of different CO gas fractions on the activity of adapted biomass was assessed in batch experiments. Formate and acetate production from CO as sole electron donor occurred, but higher formate formation and lower acetate formation were observed with increasing CO fractions (Figure 5). Formate is produced by some bacteria and archaea that grow acetogenically on CO, such as *Clostridium ljungdahlii*, *Methanosarcina acetivorans*, and *Archaeoglobus fulgidus* [40,41,42]. Currently, *Fuchsiella alkaliacetigena* and *F. ferrireducens* are the only isolates representing haloalkaliphilic hydrogenotrophic acetogen, but both are unable to grow in the presence of CO [43,44]. Bacteria belonging to Clostridiaceae family dominated the microbial community in the bioreactor (Figure 6). With further analysis, ~99% of these sequences were closely related to the *Tindallia* genus. *Tindallia* related bacteria were previously detected as dominant bacteria in H_2_ fed sulfate and thiosulfate-reducing bioreactors and their role in formate production was hypothesized [7,8,23]. *Tindallia* related bacteria were also detected in high relative abundance in a 12 L bench-scale haloalkaliphilic bioreactor that was continuously operated for the treatment of high concentrations of sulfate [45]. The bioreactor was fed with glucose as substrate, that was first converted into ethanol, lactate, acetate and formate in the lower region of the bioreactor by fermentative microbes, and then sulfate reducers metabolized the organic acids coupled to sulfate reduction, both in the lower as well as in the higher regions of the bioreactor. Formate was detected in all regions of the bioreactor as well as in the effluent. Despite the low energy yield of H_2_-driven formate production (Equation (7), Table 1), one of the *Tindallia* isolates was capable of H_2_-driven formate production coupled to growth [46]. As *Tindallia* related bacteria were dominant during the whole bioreactor operation period, it was likely that they had adapted to CO.

CO-oxidation coupled to sulfate reduction at pH neutral, thermophilic conditions by mixed or pure cultures, such as *Desulfotomaculum carboxydivorans* and *Archaeoglobus fulgidus*, is well documented [20,21,22,40,47,48]. In the present study, sulfate reduction only started after the removal of CO and no CO-driven sulfate reduction was detected in batch experiments (Figure 7). This suggested that hydrogen and/or formate were the electron donors for sulfate reduction. Interestingly, incubations with CO and 50 mM formate did show sulfate reduction. This indicated that formate could act as an electron donor for sulfate reduction in the presence of CO, possibly circumventing hydrogenases. The sulfate reducers detected in both the suspended and attached biomass were related to the *Desulfohalobiaceae* family (Figure 6). Zooming in to genus level of *Desulfohalobiaceae* related sequences detected in the libraries, all were closely related to *Desulfonatronovibrio*. The studied isolates belonging to this genus can use both H_2_ and formate to reduce sulfate [49]. However, to date there is no information available on the effect of CO on these bacteria and how formate can be used as an electron donor in the presence of CO. 

### 3.4. Biomass Aggregation

The addition of CO to the bioreactor influenced the formation of biomass aggregates that were not attached to sand. After addition of CO, small compact biomass aggregates were observed while no biofilm formation on sand particles was detected (Appendix A). A similar effect was previously reported in bioreactors operated at neutral conditions [22]. The formation of biomass aggregates under haloalkaline conditions was previously noted in a bioreactor fed with H_2_, but their appearance was not as compact as observed with CO (Appendix A) [23]. The particles in the bioreactor were mainly dominated by biomass aggregates that decreased in diameter upon CO addition (Figure 4). The maximum size achieved after CO addition did not increase with time as can be seen by the similar particle size distribution at days 125 and 216 (Figure 4). This phenomenon might be related to the haloalkaline conditions and more specifically, the effect of pH on the hydrophobicity of cells surfaces or absence of soluble divalent cations for EPS stabilization [50,51,52].

The observed aggregates were more compact than those in a bioreactor without addition of CO. This might be connected to a metabolic relationship between CO oxidizers and sulfate reducers as was observed for pH neutral conditions [22]. Focusing on the microbial composition of the aggregates in settled biomass on day 123; ~58% consisted of *Desulfohalobiaceae* and more specifically *Desulfonatronovibrio*-like bacteria (Figure 6). All studied *Desulfonatronovibrio* sp. isolates use acetate as carbon source for growth [49]. Thus, aggregation of sulfate reducers together with CO-oxidizing microbes, such as acetogens, could have enhanced their growth due to acetate production. Additionally, the lower in situ CO concentrations generated by CO oxidizing acetogens might also have been more favorable to the sulfate reducers.

### 3.5. Application in Gas Biodesulfurization

Syngas can be used as electron donor for sulfate and thiosulfate-reducing bioreactors operated under haloalkaline conditions. However, the CO concentration in the syngas varies considerably depending on the feedstock used and production method, from 0 up to higher than 50% [53,54]. Even though inhibition effects were observed during the spike of 60% CO performed in this study, such spikes can be controlled by a proper production of syngas or pre-treatment of syngas [54]. Thus, despite using syngas without removing CO would decrease the cost of the electron donor, the capacity of biomass to withstand CO fractions only up to 15% might make the application of bleed stream treatment using syngas and recycling an interesting option for biodesulfurization systems (Figure 8). 

Acetate production might be a drawback for the application of such bioreactor in biodesulfurization systems. For the sulfate and thiosulfate reducing bioreactor, excessive acetate production consumes additional electron donor, and lowers the pH. This implies the requirement of extra syngas and alkalinity to compensate for acetate production, increasing the costs of operation. After treatment, the sulfide- and acetate-rich stream from the sulfate/thiosulfate-reducing bioreactor (Figure 8, n^o^ 4) could be recycled back into the sulfide-oxidizing bioreactor (Figure 8, n^o^ 2) [55]. In the sulfide-oxidizing bioreactor, acetate leads to organic contamination of the system resulting in growth of unwanted, heterotrophic microorganisms. Acetate oxidation requires extra consumption of O_2_, increasing aeration costs, and production of CO_2_, which decreases the pH. The pH decrease affects the sulfide absorption process (Figure 8, n^o^ 1) that increases the caustic required to increase pH. 

## 4. Conclusions

Syngas, containing up to 15% CO, can be used as electron donor for a sulfate/thiosulfate-reducing bioreactor operated at haloalkaline conditions. Adaptation of the biomass to CO is required as it inhibits hydrogen-dependent microbial processes, such as sulfate reduction, formate production, and acetate production in non-adapted biomass. After adaptation, the biomass sulfate/thiosulfate reduction activity was comparable to previous studies using other electron donors, such as H_2_ or formate. Acetate production was the dominant conversion in the bioreactor when CO was supplied. Acetate production seemed to enhance the growth of sulfate-reducing bacteria that require acetate as the carbon source. The high acetate concentration in the treated bleed stream has consequences for the sulfide oxidation step and sulfide absorption step of the biodesulfurization process after it is recycled. 

## Figures and Tables

**Figure 1 microorganisms-08-01451-f001:**
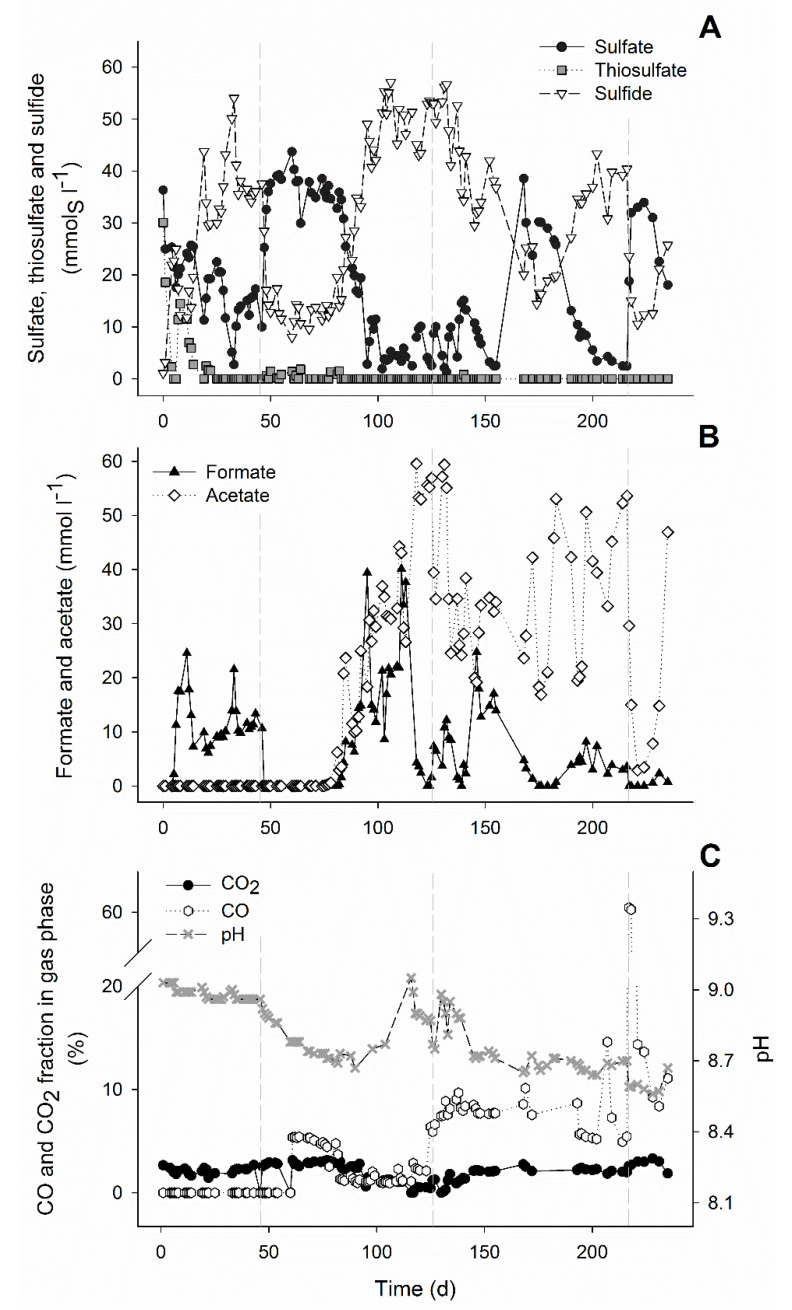
Concentrations of sulfate, thiosulfate and sulfide (**A**), formate and acetate (**B**), and pH and fractions of CO and CO_2_ in the gas phase of the bioreactor during the bioreactor experiment (**C**), Vertical dashed lines represent the start of CO experiments: 5% CO (1st), 15% CO (2nd), and 60% CO spike (3rd).

**Figure 2 microorganisms-08-01451-f002:**
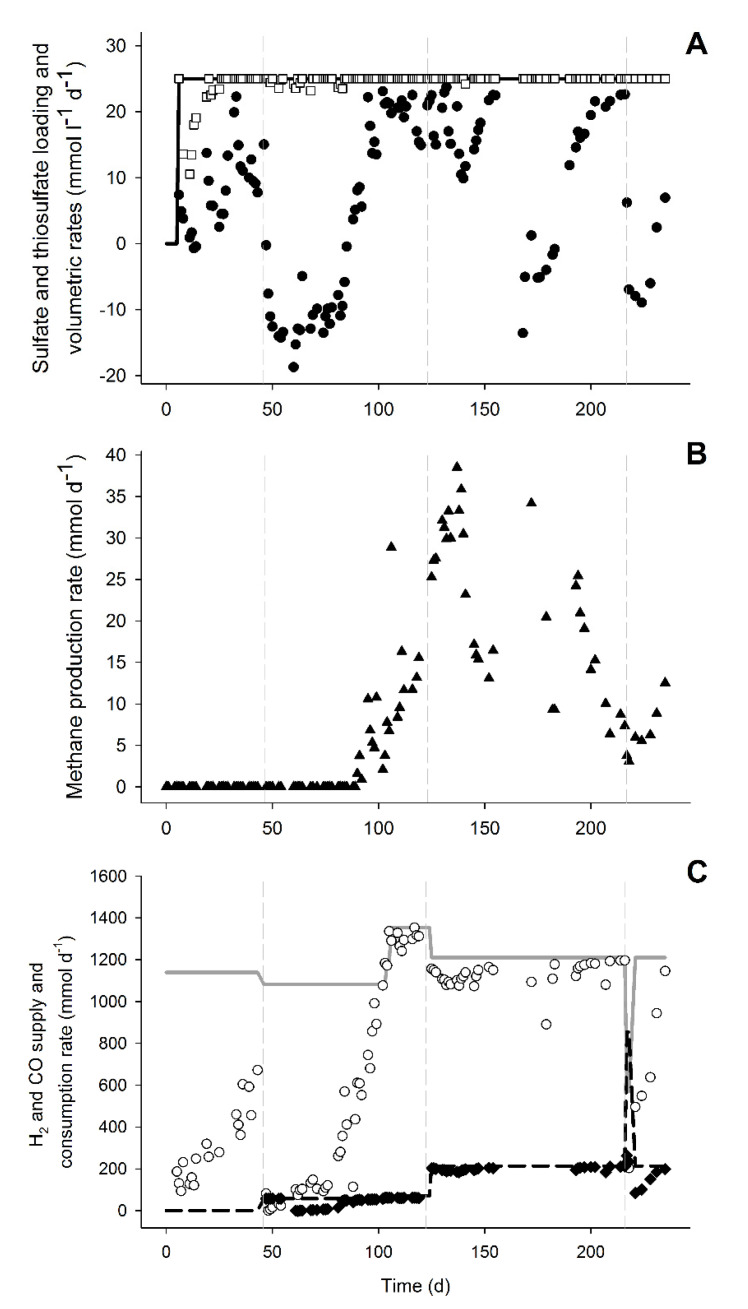
Metabolic rates and electron donor consumption during the bioreactor experiment. (**A**) Sulfate and thiosulfate loading rate (represented by the continuous line), sulfate (●), thiosulfate (□), and volumetric reduction rates (r_vs_). (**B**) Total methane production rate (r_vp_). (**C**) Supply of H_2_ (continuous line) and CO (dashed line) along with consumption rate of H_2_ (○) and CO (♦). Vertical dashed lines represent the start of CO addition: 5% CO (1st), 15% CO (2nd), and 60% CO spike (3rd).

**Figure 3 microorganisms-08-01451-f003:**
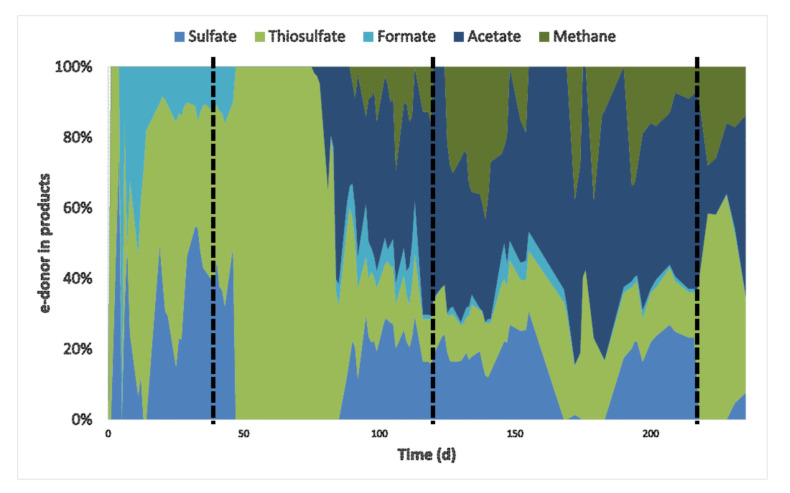
Fraction of electron donor (H_2_ + CO) used by the different metabolisms: sulfate and thiosulfate reduction and formate, acetate and methane production. For calculation, all values of electron donors and acceptors were converted to mmol/d. Vertical dashed lines represent the beginning of CO experiments: 5% CO (1st), 15% CO (2nd), and 60% CO spike (3rd).

**Figure 4 microorganisms-08-01451-f004:**
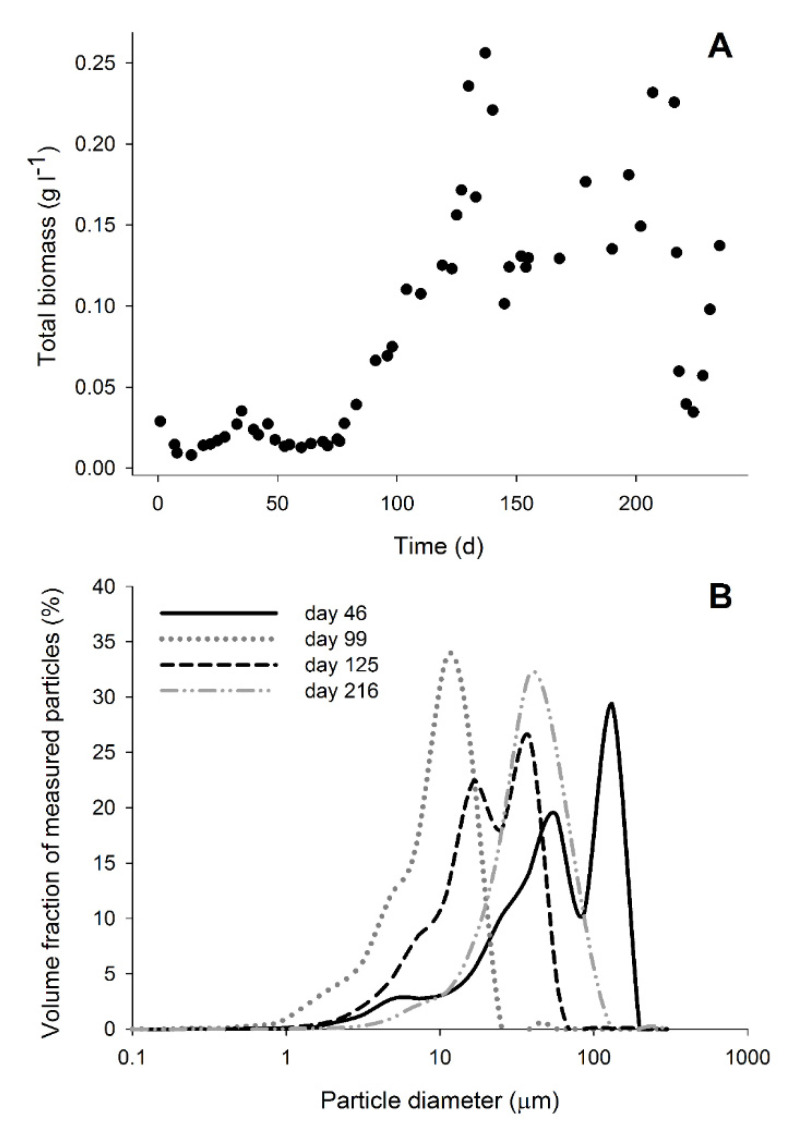
Biomass in the bioreactor. (**A**) Total biomass concentration (C_tx_) in the bioreactor during its operation. (**B**) Particle diameter distribution of the biomass on days 46 (without CO), 99 (5% CO), 125 (just after increase to 15%), and 216 (stable phase with 15% CO).

**Figure 5 microorganisms-08-01451-f005:**
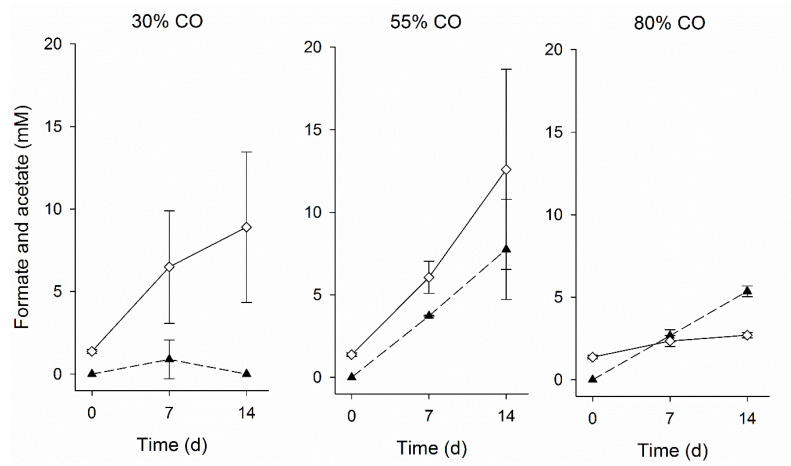
Formate (▲) and acetate (◊) production during the batch tests with biomass from day 54 incubated with 30%, 55%, and 80% CO in the gas phase. The vertical bars indicate the standard deviation among triplicates.

**Figure 6 microorganisms-08-01451-f006:**
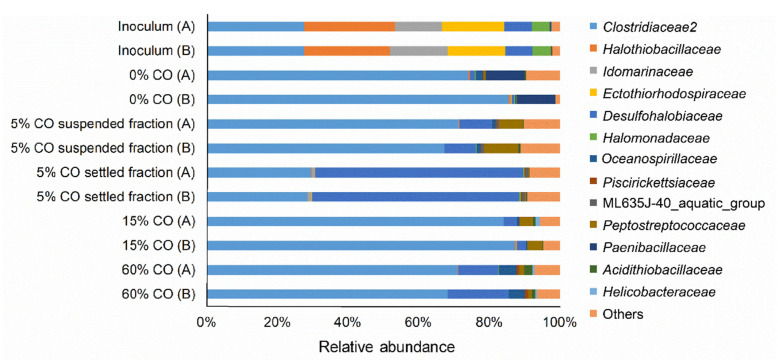
Microbial 16S rRNA relative abundance on family level in the inoculum and biomass samples from the end of the runs without CO (day 46), with 5% CO (day 123), with 15% CO (day 216), and after the spike with 60% CO (day 218). For the samples of day 123, the suspended and settling fractions were separated and analyzed separately. A and B represent duplicates for the corresponding day. OTUs with less than 0.5% relative abundance were grouped in “others.”

**Figure 7 microorganisms-08-01451-f007:**
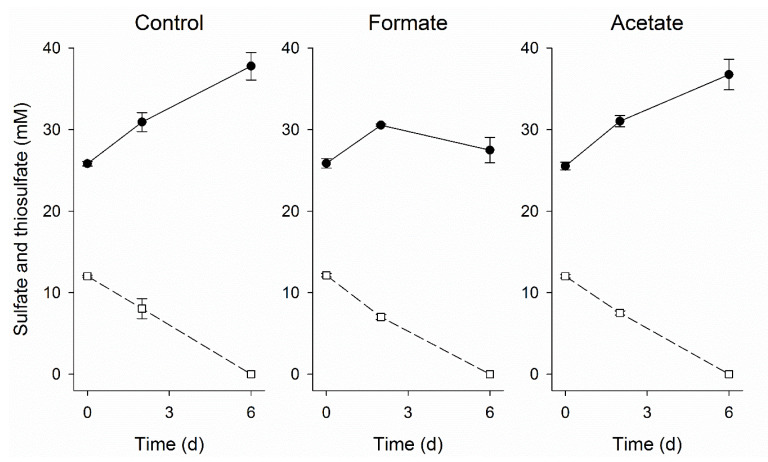
Sulfate (●) and thiosulfate (□) conversion during the batch tests with biomass from day 90 incubated with 25% CO in the gas phase. The control test was performed with regular medium while 25 mM formate and 12.5 mM acetate were added to the formate and acetate experiments, respectively. Vertical bars represent the standard deviation among triplicates.

**Figure 8 microorganisms-08-01451-f008:**
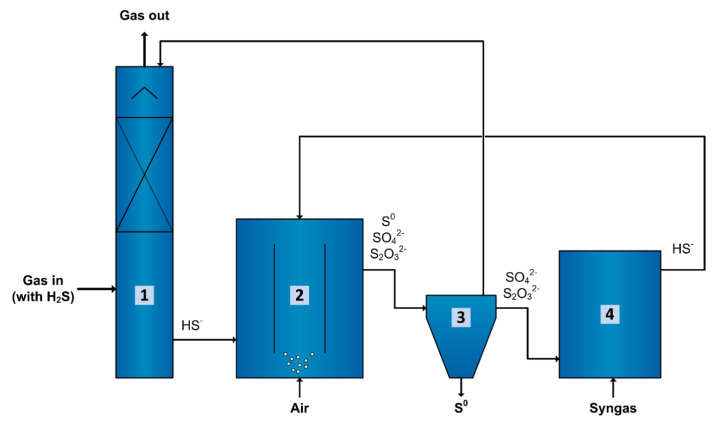
Biological process for gas desulfurization using an anaerobic sulfate/thiosulfate reducing bioreactor to enable bleed stream recycling. The sulfide (H_2_S) present in the gas is dissolved in an alkaline solution as HS^−^ using a scrubber (1). This HS^−^ rich solution goes to an aerobic bioreactor (2) where it is biologically oxidized under controlled microaerophilic conditions to mostly elemental sulfur (S^0^) and a minor fraction to sulfate (SO_4_^2−^) followed by chemical oxidation via polysulfides, to thiosulfate (S_2_O_3_^2−^). The S^0^ is separated in a settler (3) and most of the liquid is recycled to the scrubber (1). Part of the liquid from the settler (3) goes to an anaerobic bioreactor (4) where SO_4_^2−^ and S_2_O_3_^2−^ are reduced to HS^−^ using syngas as an electron donor. The HS^−^ produced is recycled back to the aerobic bioreactor (2). With time, this prevents accumulation of SO_4_^2−^ and S_2_O_3_^2−^ in the whole system and theoretically reduces the amount of caustic required to increase the pH to almost zero. Additionally, it will prevent the disposal of bleed stream into the environment and maximize the S^0^ production.

**Table 1 microorganisms-08-01451-t001:** Gibbs free energy under the bioreactor actual conditions (ΔG’) for different CO and H_2_ consuming reactions at starting and continuous operation conditions.

Reaction	ΔG’ Starting (kJ mol^−1^)	ΔG’ Continuous Operation (kJ mol^−1^)	Equation No.
CO + 2H_2_O → HCO_3_^−^ + H_2_ + H^+^	−22.7 (±10.5)	−23.1 (±10.5)	(1)
4CO + 4H_2_O → C_2_H_3_O_2_^−^ + 2HCO_3_^−^ + 3H^+^	−199.6 (±26.6)	−182.9 (±26.6)	(2)
4H_2_ + 2HCO_3_^−^ + H^+^ → C_2_H_3_O_2_^−^ + 4H_2_O	−108.9 (±26.9)	−90.5 (±26.9)	(3)
4CO + SO_4_^2−^ + 4H_2_O → 4HCO_3_^−^ + HS^−^ + 3H^+^	−240.6 (±34.3)	−218.8 (±34.3)	(4)
4H_2_ + SO_4_^2−^ + H^+^ → HS^−^ + 4H_2_O	−150.0 (±25.0)	−126.4 (±25.0)	(5)
CO + H_2_O → HCO_2_^−^ + H^+^	−38.0 (±6.5)	−32.3 (±6.5)	(6)
H_2_ + HCO_3_^−^ → HCO_2_^−^ + H_2_O	−15.2 (±8.9)	−9.2 (±8.9)	(7)
4CO + 5H_2_O → CH_4_ + 3HCO_3_^−^ + 3H^+^	−215.4 (±30.6)	−198 (±30.6)	(8)
4H_2_ + HCO_3_^−^ + H^+^ → CH_4_ + 3H_2_O	−124.4 (±25.1)	−105.6 (±25.1)	(9)

Conditions used for calculations: pH 9, 1.5 ionic strength, 825 mM HCO_3_^−^. Starting conditions: 50 mM sulfate, 1 mM sulfide, 1 mM acetate, 1 mM formate, 150 mbar CO, 850 mbar H_2_, 10 mbar methane. Continuous operation conditions (average of days 190 to 216): 5.7 mM sulfate, 36.9 mM sulfide, 37.8 mM acetate, 4.5 mM formate, 68 mbar CO, 333 mbar H_2_, 575 mbar methane. Error for each ΔG’ estimation is presented in brackets because formation energy of the compounds under high pH and ionic strength are estimated based on models.

**Table 2 microorganisms-08-01451-t002:** Overview of operational characteristics of bioreactor runs.

	Period (Days)	Mode
Start-up	0–6	Batch
0% CO	7–45	Continuous
5% CO	46–123	Continuous
15% CO	124–216	Continuous
60% CO spike *	217–235	Continuous

* See experimental design for a detailed description.

**Table 3 microorganisms-08-01451-t003:** Conditions and performance of sulfate and thiosulfate reducing bioreactors operated under haloalkaline conditions.

	This Study	Sousa et al. 2020	Zhou and Xing 2015	Sousa et al. 2017
Reactor type	Gas lift with 3 phase separator	Gas lift with 3 phase separator	Anaerobic filter	Gas lift with 3 phase separator
e^−^ acceptor	Sulfate/thiosulfate	Sulfate/thiosulfate	Sulfate	Thiosulfate
e^−^ donor	H_2_/CO	H_2_	Formate	H_2_
pH	9	9	9.5	9
Na^+^ conc. (M)	1.5	1.5	1	1.5
Temperature (°C)	35	35	37	35
HRT (d)	1	1	1	1.7
Max loading rate (mmol_S_ L^−1^ d^−1^)	50	100	88.5	50
Sulfidogenic rate (mmol_S_ L^−1^ d^−1^)	46.8 (±0.8)	85.45 (±3)	85.05 (±0.2)	28.7 (±0.8)
Side products	Formate/Acetate/Methane	Formate/Acetate/Methane	Acetate	Formate
Biomass conc. (mg L^−1^)	197 (±39)	127 (±41)	N.D.	14 (±2.2)

N.D.—No data available.

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
