# Peer review of "Syngas as Electron Donor for Sulfate and Thiosulfate Reducing Haloalkaliphilic Microorganisms in a Gas-Lift Bioreactor"

_microorganisms, 2020, doi:10.3390/microorganisms8091451_

Round 1

Reviewer 1 Report

General comments

The manuscript reported the evidence for using syngas as an electron donor for sulfate and thiosulfate reduction by haloalkaliphilic bacteria in a gas-lift bioreactor. The application in gas biodesulfurization with advantages and disadvantages was drawn out. The overall manuscript was well written and understandable. Some minor point should be revised before publication.

Specific comments

- Line 168: Please check the format.

- How was the data was collected in Fig. 1? No standard deviation (error bars) was observed.

- Line 338: It should be "Relative abundance of bacterial families in the..."

- Fig. 6: bacterial family name should be italicized. Check the spelling of "Clostridiaceae2". Do not use an unidentified expression such as "ML635J-40_aquatic group". If the family level is unclassified, the expression "unclassified + upper-level name such as order/ class/ phylum name" could be used. If no taxa level is identified, "unclassified bacteria" could be used.

- Line 350: It should be "Bacteria belonging to Clostridiaceae family dominated...."

Author Response

General comments

The manuscript reported the evidence for using syngas as an electron donor for sulfate and thiosulfate reduction by haloalkaliphilic bacteria in a gas-lift bioreactor. The application in gas biodesulfurization with advantages and disadvantages was drawn out. The overall manuscript was well written and understandable. Some minor point should be revised before publication.

Re: thank you for your appreciation of our work

Specific comments

- Line 168: Please check the format.

Re: we have revised the format

- How was the data was collected in Fig. 1? No standard deviation (error bars) was observed.

Re: Legend of Figure 1. Concentrations of sulfate, thiosulfate and sulfide (A), formate and acetate (B), and pH and fractions of CO and CO2 in the gas phase of the bioreactor during the bioreactor experiment (C). Vertical dashed lines represent the start of CO experiments: 5% CO (1st), 15% CO (2nd), and 60% CO spike (3rd).

The results presented are daily measurements of the reactor’s performance which are measured routinely, and not in replicates. Details are presented in the materials and methods section paragraph 1.6.

- Line 338: It should be "Relative abundance of bacterial families in the..."

Re: Legend of Figure 6 provides the complete details: “Microbial 16S rRNA relative abundance on family level in the inoculum and biomass samples from the end of the runs without CO (day 46), with 5% CO (day 123), with 15% CO (day 216), and after the spike with 60% CO (day 218). For the samples of day 123, the suspended and settling fractions were separated and analyzed separately. A and B represent duplicates for the corresponding day. OTUs with less than 0.5% relative abundance were grouped in “others”.

Therefor we kept the line 345 (old line338) in the figure as it was

- Fig. 6: bacterial family name should be italicized. Check the spelling of "Clostridiaceae2". Do not use an unidentified expression such as "ML635J-40_aquatic group". If the family level is unclassified, the expression "unclassified + upper-level name such as order/ class/ phylum name" could be used. If no taxa level is identified, "unclassified bacteria" could be used.

Re: we have kept the "ML635J-40_aquatic group” as it is, since it is useful information to zoom in to the aquatic group and retrieve phylogenetic or ecological information.

We have modified figure 6 according to your suggestions

- Line 350: It should be "Bacteria belonging to Clostridiaceae family dominated...."

Re: We have revised the text accordingly

Reviewer 2 Report

The authors studied the use of syngas (H2/CO mixture) instead of pure H2 in the biodesulfurization process as a strategy for cost reduction. The research topic is interesting and in the scope of the journal and the manuscript is well written. The research methodology is correct and the results and discussion are comprehensive. In my opinion, the manuscript could be published in Microorganism if the following comments are addressed.

comments:

1-The authors need to provide more information in the abstract on the merit of the suggested method in comparison with the same process with pure H2.

2-Please improve the quality of Fig.3. The texts are not readable. 

3-As shown in Table3, the sulfidogenic rate is lower compared to the previous studies. The authors need to discuss this issue in more detail.

Author Response

The authors studied the use of syngas (H2/CO mixture) instead of pure H2 in the biodesulfurization process as a strategy for cost reduction. The research topic is interesting and in the scope of the journal and the manuscript is well written. The research methodology is correct and the results and discussion are comprehensive. In my opinion, the manuscript could be published in Microorganism if the following comments are addressed.

Re: thank you for your appreciation of our work

comments:

1-The authors need to provide more information in the abstract on the merit of the suggested method in comparison with the same process with pure H2.

Re: we have added the following sentence to the abstract to emphasize the merit of using syngas: “Syngas is produced in the gas reforming process and consists mainly of H2, carbon monoxide (CO), and carbon dioxide (CO2). Purification of syngas to obtain pure H2 implies higher costs due to additional post-treatment. Therefore, the use of syngas has merit in the biodesulfurization process.

2-Please improve the quality of Fig.3. The texts are not readable. 

Re: we have increased the fonts in Figure 3

3-As shown in Table3, the sulfidogenic rate is lower compared to the previous studies. The authors need to discuss this issue in more detail.

Re: we have added to the discussion (line 261): “Moreover, increasing the length of the different stages may contribute to a further increase in the sulfidogenic rates.”